# Multi-agent Trajectory Prediction with Fuzzy Query Attention

**Nitin Kamra**
Department of Computer Science
University of Southern California
Los Angeles, CA, USA
nkamra@usc.edu

**Hao Zhu**
Department of Computer Science, School of EECS
Peking University
Beijing, China
hzhu1998@pku.edu.cn

**Dweep Trivedi**
Department of Computer Science
University of Southern California
Los Angeles, CA, USA
dtrivedi@usc.edu

**Ming Zhang**
Department of Computer Science, School of EECS
Peking University
Beijing, China
mzhang_cs@pku.edu.cn

**Yan Liu**
Department of Computer Science
University of Southern California
Los Angeles, CA, USA
yanliu.cs@usc.edu

## Abstract

Trajectory prediction for scenes with multiple agents and entities is a challenging problem in numerous domains such as traffic prediction, pedestrian tracking and path planning. We present a general architecture to address this challenge which models the crucial inductive biases of motion, namely, inertia, relative motion, intents and interactions. Specifically, we propose a relational model to flexibly model interactions between agents in diverse environments. Since it is well-known that human decision making is fuzzy by nature, at the core of our model lies a novel attention mechanism which models interactions by making continuous-valued (fuzzy) decisions and learning the corresponding responses. Our architecture demonstrates significant performance gains over existing state-of-the-art predictive models in diverse domains such as human crowd trajectories, US freeway traffic, NBA sports data and physics datasets. We also present ablations and augmentations to understand the decision-making process and the source of gains in our model.

## 1  Introduction

Multi-agent settings are ubiquitous and predicting trajectories of agents in motion is a key challenge in many domains, e.g., traffic prediction [28, 15], pedestrian tracking [1, 3] and path planning [20]. In order to model multi-agent settings with complex underlying interactions, several recent works based on graphs and graph neural networks have achieved significant success in prediction performance [23, 14]. However, modeling interactions between two agents is challenging because it is not a binary true/false variable but is rather fuzzy[1] by nature. For instance, a person driving a car on a freeway

might reason along these lines: "The car in front of me is slowing down so I should also step on the brake lightly to avoid tailing the car closely", wherein the decisions *slowing down*, *braking lightly* and *tailing closely* are all continuous-valued in nature. Since such fuzzy representations enter routinely into human interactions and decision making, we posit that learning to predict trajectories of interacting agents can benefit from fuzzy (continuous-valued) decision making capabilities.

Motivated by this observation, we present a novel Fuzzy Query Attention (FQA) mechanism to solve the aforementioned challenges. FQA models pairwise attention to decide about when two agents are interacting by learning keys and queries which are combined with a dot-product structure to make continuous-valued (fuzzy) decisions. It also simultaneously learns how the agent under focus is affected by the influencing agent given the fuzzy decisions. We demonstrate significant performance gains over existing state-of-the-art predictive models in several domains: (a) trajectories of human crowd, (b) US freeway traffic, (c) object motion and collisions governed by Newtonian mechanics, (d) motion of charged particles under electrostatic fields, and (e) basketball player trajectories, thereby showing that FQA can learn to model very diverse kinds of interactions. Our experiments show that the fuzzy decisions made over time are highly predictive of interactions even when all other input features are ignored. Our architecture also supports adding human knowledge in the form of fuzzy decisions, which can provide further gains in prediction performance.

## 2   Related work

Multi-agent trajectory prediction is a well-studied problem spanning across many domains such as modeling human interactions for navigation, pedestrian trajectory prediction, spatio-temporal prediction, multi-robot path planning, traffic prediction, etc. Early work on predicting trajectories of multiple interacting agents dates back to more than two decades starting from Helbing and Molnar's social force model [10] and its later extensions [19, 28] aimed at modeling behavior of humans in crowds, pedestrians on highways and vehicles on highways and freeways. Since a comprehensive review of all domains is out of scope of this work, we only survey some of the most recent literature.

Due to the growing success being enjoyed by deep recurrent models like RNNs and LSTMs in sequence prediction, recurrent models with LSTM-based interaction modeling have recently become predominant for multi-agent trajectory prediction [17]. To aggregate influence of multiple interactions, various pooling mechanisms have been proposed for both human crowds modeling [1, 8] and for predicting future motion paths of vehicles from their past trajectories [6]. Many state-of-the-art models have also incorporated attention mechanisms to predict motion of human pedestrians in crowds [24, 27, 7]. For a review and benchmark of different approaches in this domain, we refer the interested reader to [3]. Many recent works have also studied trajectory prediction for particles in mechanical and dynamical systems [4, 14, 18], for predicting trajectories of soccer and basketball players [30, 11, 22, 29] and for predicting trajectories in multi-robot path planning [20].

A recurring theme in the above works is to view the agents/entities as nodes in a graph while capturing their interactions via the graph edges. Since graph neural networks can be employed to learn patterns from graph-structured data [2], the problem reduces to learning an appropriate variant of graph neural networks to learn the interactions and predict the trajectories of all agents [23]. Recent works have devised different variants of graph networks, e.g. with direct edge-feature aggregation [9, 2, 21], edge-type inference [14], modeling spatio-temporal relations [12], and attention on edges between agents [26] to predict multi-agent trajectories in diverse settings.

Our work assumes a graph-based representation for agents but differs from above literature in its novel attention mechanism to capture interactions between agents. Our attention mechanism learns to make continuous-valued decisions which are highly predictive of when and how two agents are interacting. It further models the effects of the interactions on agents by learning appropriate responses for these decisions and outperforms state-of-the-art methods in modeling multi-agent interactions.

## 3   Fuzzy Query Attention model

**Problem formulation**: Following previous work [1, 14], we assume a given scene which has been pre-processed to obtain the spatial coordinates $p_i^t = (x_i^t, y_i^t)$ of all agents $i \in 1 : N$ at a sequence of time-steps $t \in 1 : T$. The task is to observe all agents from time 1 to $T_{obs}$, infer their motion characteristics and ongoing interactions and predict their positions for time-steps $T_{obs} + 1$ to $T$. In all

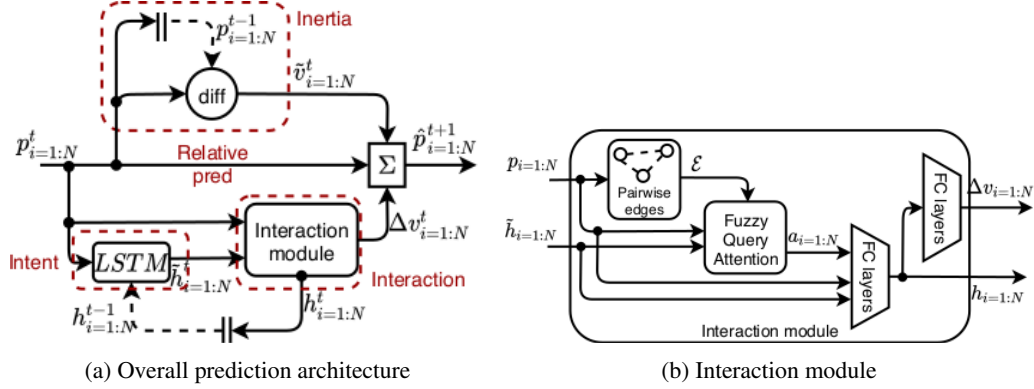

|  (a) Overall prediction architecture  |  (b) Interaction module  |

Figure 1: Multi-agent prediction architecture using Fuzzy Query Attention at time $t$: (a) Overall architecture takes positions ($p$) of all agents, computes a first-order estimate of velocity ($\tilde{v}$) and incorporates effects of interactions between agents via a correction term ($\Delta v$) thereby predicting the positions at the next time-step ($\hat{p}^{t+1}$); (b) the Interaction module generates pairwise edges between agents ($\mathcal{E}$) and uses the FQA module to account for interactions and generate the aggregate effect ($a$) for each agent which is used to update their LSTM state ($h$) and predict the velocity correction ($\Delta v$).

subsequent text, $p^t = \{p_1^t, p_2^t, \ldots, p_N^t\}$ represents the set of positions of all agents at time $t$, while $p_i = [p_i^1, p_i^2, \ldots, p_i^T]$ represents the sequence of positions of a single agent $i$ at all time-steps. $v$ is used to denote velocity, tilde symbol ($\tilde{\cdot}$) on the top to denote intermediate variables and hat symbol ($\hat{\cdot}$) on the top for predicted quantities or unit vectors (will be clear from context).

**Design principles**: We present our architecture which incorporates the following crucial inductive biases required for motion prediction:

- **Inertia**: Most inanimate entities move with constant velocity until acted upon by external forces. This also acts as a good first-order approximation for animate agents for short time-intervals, e.g., pedestrians walk with nearly constant velocities unless they need to turn or slow down to avoid collisions.
- **Motion is relative**: Since motion between two agents is relative, one should use agents' relative positions and velocities while predicting future trajectories (*relative observations*) and should further make predictions as offsets relative to the agents' current positions (*relative predictions*).
- **Intent**: Unlike inanimate entities, animate agents have their own intentions which can cause deviations from inertia and need to be accounted for in a predictive model.
- **Interactions**: Both inanimate and animate agents can deviate from their intended motion due to influence by other agents around them and such interaction needs to be explicitly modeled.

**Prediction architecture**: The overall prediction architecture (Figure 1a) takes the spatial positions of all agents i.e. $p_{i=1:N}^t$ as input at time $t$. We use the observed positions for $t \leq T_{obs}$ and the architecture's own predictions from the previous time-step for $t > T_{obs}$. We predict each agent's position at the next time-step $\hat{p}_i^{t+1}$ as an offset from its current position $p_i^t$ to capture the *relative prediction* inductive bias. We further break each offset into a first-order constant velocity estimate $\tilde{v}_i^t$ which accounts for the *inertia* inductive bias and a velocity correction term $\Delta v_i^t$ which captures agents' *intents* and inter-agent *interactions* (see eq 1). The first-order estimate of velocity ($\tilde{v}_i^t$) is made by a direct difference of agents' positions from consecutive time steps (eq 2). To capture agents' intents, an LSTM module is used to maintain the hidden state ($h_i^{t-1}$) containing the past trajectory information for the $i^{th}$ agent. The learnable weights of the LSTM are shared by all agents. To compute the correction term ($\Delta v_i^t$), a preliminary update is first made to the LSTM's hidden state using the incoming observation for each agent. This preliminary update captures the deviations from inertia due to an agent's own intentional acceleration or retardation (eq 3). The intermediate hidden states $\tilde{h}_i^t$ and the current positions of all agents are further used to infer the ongoing interactions between agents, aggregate their effects and update the hidden state of each agent to $h_i^t$ while also

computing the correction term for the agent's velocity via an interaction module (eq 4).

$$\hat{p}_i^{t+1} = p_i^t + \tilde{v}_i^t + \Delta v_i^t, \qquad \forall i \in 1:N \tag{1}$$

$$\text{(Inertia): } \tilde{v}_i^t = p_i^t - p_i^{t-1}, \qquad \forall i \in 1:N \tag{2}$$

$$\text{(Agent's Intents): } \tilde{h}_i^t = \text{LSTM}(p_i^t, h_i^{t-1}), \ \ \forall i \in 1:N \tag{3}$$

$$\text{(Interactions): } h^t, \Delta v^t = \text{InteractionModule}(p^t, \tilde{h}^t) \tag{4}$$

Since computation in all sub-modules happens at time $t$, we drop the superscript $t$ from here on.

**Interaction module**: The interaction module (Figure 1b) first creates a graph by generating directed edges between all pairs of agents (ignoring self-edges)[2]. The edge set $\mathcal{E}$, the positions and the states of all agents are used to compute an attention vector $a_i$ for each agent aggregating all its interactions with other agents via the Fuzzy Query Attention (FQA) module (eq 5). This aggregated attention along with each agent's current position and intermediate hidden state is processed by subsequent fully-connected layers to generate the updated state $h_i$ (which is fed back into the LSTM) and the velocity correction $\Delta v_i$ for each agent (eqs 6 and 7).

$$a = \text{FQA}(p, \tilde{h}, \mathcal{E}) \tag{5}$$

$$h_i = FC_2(ReLU(FC_1(p_i, h_i, a_i))), \quad \forall i \in 1:N \tag{6}$$

$$\Delta v_i = FC_4(ReLU(FC_3(h_i))), \qquad \forall i \in 1:N \tag{7}$$

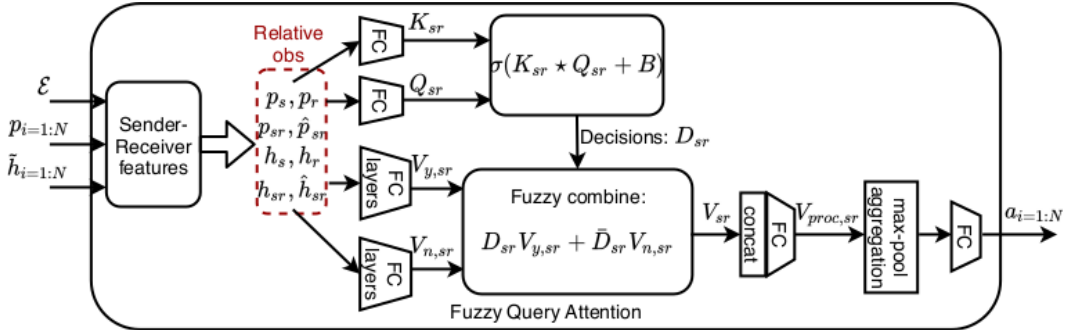

Figure 2: FQA module generates keys ($K_{sr}$), queries ($Q_{sr}$) and responses ($V_{y,sr}, V_{n,sr}$) from sender-receiver features between agent pairs, combines the responses according to the fuzzy decisions ($D_{sr}$), and aggregates the concatenated responses into a vector ($a$) per agent.

**Fuzzy Query Attention**: The FQA module views the graph edges as sender-receiver ($s - r$) pairs of agents. At a high level, it models the aggregate effect of the influence from all sender agents onto a specific receiver agent (Figure 2). To do so, we build upon the key-query-value based self-attention networks introduced by Vaswani *et al.* [25]. FQA first generates independent features: $p_s, p_r, h_s$ and $h_r$ for the senders and receivers by replicating $p$ and $h$ along each edge. It also generates relative features: $p_{sr} = p_s - p_r$ (relative displacement), $h_{sr} = h_s - h_r$ (relative state), $\hat{p}_{sr} = p_{sr}/\|p_{sr}\|$ (unit-vector along $p_{sr}$) and $\hat{h}_{sr} = h_{sr}/\|h_{sr}\|$ (unit-vector along $h_{sr}$) to capture the *relative observations* inductive bias. These features $f_{sr} = \{p_s, p_r, p_{sr}, \hat{p}_{sr}, h_s, h_r, h_{sr}, \hat{h}_{sr}\}$ are combined by single fully-connected layers to generate $n$ keys $K_{sr} \in \mathbb{R}^{n \times d}$ and queries $Q_{sr} \in \mathbb{R}^{n \times d}$ of dimension $d$ each for every $s - r$ pair (eqs 8 and 9), which are then combined via a variant of dot-product attention to generate fuzzy[3] decisions $D_{sr} \in \mathbb{R}^n$ (eq 10):

$$K_{sr} = FC_5(f_{sr}^\perp), \qquad\qquad\qquad \forall(s,r) \in 1:N, s \neq r \tag{8}$$

$$Q_{sr} = FC_6(f_{sr}^\perp), \qquad\qquad\qquad \forall(s,r) \in 1:N, s \neq r \tag{9}$$

$$D_{sr} = \sigma(K_{sr} \star Q_{sr} + B) = \sigma\left(\sum_{dim=1} K_{sr} \odot Q_{sr} + B\right), \qquad \forall(s,r) \in 1:N, s \neq r \tag{10}$$

where $\odot$ represents element-wise product, $B \in \mathbb{R}^n$ is a learnable bias parameter, $\sigma$ stands for the sigmoid activation function and $\perp$ stands for the detach operator[4]. As a consequence of this formulation, $D_{sr} \in [0,1]^n$ can be interpreted as a set of $n$ continuous-valued decisions capturing the interaction between agents $s$ and $r$. These can now be used to select the receiving agent's response to the current state of the sending agent. For this, the sender-receiver features are parsed in parallel by two-layer neural networks (with the first layer having a ReLU activation) to generate yes-no responses $V_{y,sr}, V_{n,sr} \in \mathbb{R}^{n \times d_v}$ corresponding to $D_{sr}$ being 1 (*yes*) or 0 (*no*) respectively (eqs 11 and 12). Though all the $s - r$ features can be used here, our preliminary experiments showed that including only a subset of features ($h_s$ and $p_{sr}$) gave comparable results and led to considerable saving in the number of parameters, so we only use this subset of features to generate the yes-no responses. These responses are then combined using a *fuzzy if-else* according to decisions $D_{sr}$ and their complements $\bar{D}_{sr} = 1 - D_{sr}$ to generate the final responses $V_{sr} \in \mathbb{R}^{n \times d_v}$ (eq 13):

$$V_{y,sr} = FC_8(ReLU(FC_7(p_{sr}, h_s))), \qquad \forall (s,r) \in 1:N, s \neq r \qquad (11)$$

$$V_{n,sr} = FC_{10}(ReLU(FC_9(p_{sr}, h_s))), \qquad \forall (s,r) \in 1:N, s \neq r \qquad (12)$$

$$\text{(Fuzzy if-else): } V_{sr} = D_{sr}V_{y,sr} + \bar{D}_{sr}V_{n,sr}, \qquad \forall (s,r) \in 1:N, s \neq r \qquad (13)$$

The $n$ final responses generated per agent pair ($\in \mathbb{R}^{n \times d_v}$) are then concatenated ($\in \mathbb{R}^{nd_v}$) and final responses from all senders are aggregated on the respected receivers by dimension-wise max-pooling to accumulate effect of all interactions on the receiver agents (eqs 14 and 15). Since max-pooling loses information while aggregating, we pre-process the final responses to increase the dimensions and retain more information followed by subsequent post-processing after aggregation to reduce the number of dimensions again (eqs 14 and 16):

$$V_{proc,sr} = FC_{11}(\text{concat}(V_{sr})) \qquad (14)$$

$$V_{proc,r} = \text{maxpool}_{s:(s-r)\in\mathcal{E}} V_{proc,sr} \qquad (15)$$

$$a_r = FC_{12}(V_{proc,r}), \qquad \forall r \in 1:N. \qquad (16)$$

**Strengths of FQA**: While originally motivated from multi-head self-attention [25], FQA differs significantly in many respects. Firstly, FQA generalizes self-attention to pairwise-attention which attends to an ordered pair (sender-receiver) of entities and captures the interaction effects of the sender on the receiver. FQA has a learnable bias $B$ to improve modeling power (explained below). Further, though the original matrix-dot-product structure of self-attention requires a large memory to fit even for regular batch sizes e.g. 32, our simpler row-wise dot-product structure fits easily on a single GPU (12GB) for all datasets, while still retaining the strong performance of the dot-product attention structure. Moreover, we learn the sender-receiver features by backpropagating only through the responses ($V_{sr}$) while features are detached to generate the keys and queries. This additionally allows us to inject human knowledge into the model via handcrafted non-learnable decisions, if such decisions are available (see experiments in section 4.3).

**What kinds of decisions can FQA learn?**: Since keys and queries are linear in the senders' and receivers' states and positions, the decision space of FQA contains many intuitive decisions important for trajectory prediction, e.g.:

1. *Proximity*: FQA can potentially learn a key-query pair to be $p_{sr}$ each and the corresponding bias as $-d_{th}^2$, then the decision $D = \sigma(p_{sr}^T p_{sr} - d_{th}^2)$ going to zero reflects if agents $s$ and $r$ are closer than distance $d_{th}$. Note that such decisions would not be possible without the learnable bias parameter $B$, hence having the bias makes FQA more flexible.
2. *Approach*: Since a part of the state $h_i$ can learn to model velocity of agents $v_i$ internally, FQA can potentially learn a key-query pair of the form $K_{sr} = v_{sr}, Q_{sr} = \hat{p}_{sr}, B = 0$ to model $D = \sigma(v_{sr}^T \hat{p}_{sr} + 0)$ which tends to 0 when the agents are directly approaching each other. While we do not force FQA to learn such human-interpretable decisions, our experiments show that the fuzzy decisions learnt by FQA are highly predictive of interactions between agents (section 4.3).

**Training**: FQA and all our other baselines are trained to minimize the mean-square error in predicting next time-step positions of all agents. Since some datasets involve agents entering and exiting the scene freely between frames, we input binary masks to all models for each agent to determine the

presence of agents in the current frame and control updates for agents accordingly (masks not shown in figures to avoid clutter). All models are trained with the Adam optimizer [13] with batch size 32 and an initial learning rate of $0.001$ decaying multiplicatively by a factor $\gamma = 0.8$ every 5 epochs. All models train for at least 50 epochs after which early stopping is enabled with a max patience of 10 epochs on validation set mean-square error and training is terminated at a maximum of 100 epochs. Since we test the models by observing $T_{obs}$ (kept at $\frac{2T}{5}$ for all datasets) time-steps and make predictions until the remaining time $T$, we followed a dynamic schedule allowing all models to see the real observations for $T_{temp}$ time-steps followed by $T - T_{temp}$ of its own last time-step predictions. During training, $T_{temp}$ is initialized to $T$ and linearly decayed by 1 every epoch until it becomes equal to $T_{obs}$. We found this dynamic burn-in schedule employed during training to improve the prediction performance for all models.

## 4   Experiments

We perform multi-agent trajectory prediction on different datasets used previously in the literature with a diverse variety of interaction characteristics[5]. For datasets with no provided splits, we follow a $70 : 15 : 15$ split for training, validation and test set scenes.

1. **ETH-UCY** [3]: A human crowds dataset with medium interaction density. We sampled about 3400 scenes at random from the dataset and set $T = 20$ following prior work [1, 8].
2. **Collisions**: Synthetic physics data with balls moving on a friction-less 2D plane, fixed circular landmarks and boundary walls. The collisions between balls preserve momentum and energy, while collisions of agents with walls or immobile landmarks only preserve energy but not momentum of moving agents. Contains about 9500 scenes with $T = 25$.
3. **NGsim** [5]: US-101 and i-80 freeway traffic data with fast moving vehicles. Since this dataset features very high agent density per scene (ranging in several thousands), we chunked the freeways with horizontal and vertical lines into sub-sections to restrict the number of vehicles in a sub-scene to less than 15. We sampled about 3500 sub-scenes from the resulting chunks and set $T = 20$.
4. **Charges** [14]: Physics data with positive and negative charges moving under other charges' electric fields and colliding with bounding walls. Contains 3600 scenes with $T = 25$ involving dense attractive and repulsive interactions.
5. **NBA** [30]: Sports dataset with basketball player trajectories. We sampled about 7500 scenes with $T = 30$. This dataset features complex goal-oriented motion heavily dictated by agents' intentions. It has been included to highlight limitations of interaction modeling approaches.

We compare our FQA architecture with state-of-the-art baselines (see appendix for architecture details and unique hyperparameters of all methods):

1. **Vanilla LSTM** [VLSTM]: An LSTM preceeded and followed by fully-connected neural network layers is used to predict the offset without considering interactions.
2. **Social LSTM** [SLSTM] [1]: Recurrent architecture which models interactions by discretizing space around each agent and aggregating neighbors' latent states via a social pooling mechanism.
3. **GraphSAGE** [GSAGE] [9]: Graph neural networks with node features to model interactions between agents. We use feature-wise max-pooling for aggregating the messages along the edges.
4. **Graph Networks** [GN] [2, 23]: Graph neural networks with node features, edge features and global features to model interactions between agents. We adapt the Encoder→RecurrentGN→Decoder architecture from [23].
5. **Neural Relational Inference** [NRI] [14]: Uses graph neural networks to model interactions between agents and additionally infers edges between agents using variational inference.
6. **Graph Attention Networks** [GAT] [26]: Follows an aggregation style similar to GraphSAGE, but weighs messages passed from all sender agents via a learnt attention mechanism.

### 4.1   Prediction results

For all models, we report the Root Mean Square Error (RMSE) between ground truth and our predictions over all predicted time steps for all agents on the test set of every dataset in Table 1. The standard deviation is computed on the test set RMSE over five independent training runs differing only in their initial random seed. Our model with $n = 8$ decisions outperforms all the state-of-the-art

Table 1: Prediction error metrics for all methods on all datasets

| Model | ETH-UCY | Collisions | NGsim | Charges | NBA |
|-------|---------|-----------|-------|---------|-----|
| VLSTM | $0.576 \pm 0.002$ | $0.245 \pm 0.001$ | $5.972 \pm 0.065$ | $0.533 \pm 0.001$ | $6.377 \pm 0.053$ |
| SLSTM | $0.690 \pm 0.013$ | $0.211 \pm 0.002$ | $6.453 \pm 0.153$ | $0.485 \pm 0.005$ | $6.246 \pm 0.048$ |
| NRI | $0.778 \pm 0.027$ | $0.254 \pm 0.002$ | $7.491 \pm 0.737$ | $0.557 \pm 0.008$ | $5.919 \pm 0.022$ |
| GN | $0.577 \pm 0.014$ | $0.234 \pm 0.001$ | $5.901 \pm 0.238$ | $0.508 \pm 0.006$ | $5.568 \pm 0.032$ |
| GSAGE | $0.590 \pm 0.011$ | $0.238 \pm 0.001$ | $5.582 \pm 0.082$ | $0.522 \pm 0.002$ | $5.657 \pm 0.018$ |
| GAT | $0.575 \pm 0.007$ | $0.237 \pm 0.001$ | $6.100 \pm 0.063$ | $0.524 \pm 0.004$ | $6.166 \pm 0.052$ |
| FQA (ours) | $\mathbf{0.540 \pm 0.006}$ | $\mathbf{0.176 \pm 0.004}$ | $\mathbf{5.071 \pm 0.186}$ | $\mathbf{0.409 \pm 0.019}$ | $\mathbf{5.449 \pm 0.039}$ |

baselines on all benchmark datasets (on many by significant margins). This shows that FQA can accurately model diverse kinds of interactions. Specifically, we observe that all models find it difficult to model sparse interactions on the Collisions data, while FQA performs significantly better with lower errors presumably due to its fuzzy decisions being strongly predictive of when two agents are interacting (more detail in section 4.3). Further, though GAT also uses an attention mechanism at the receiver agents to aggregate messages, FQA outperforms GAT on all datasets showing a stronger inductive bias towards modeling multi-agent interactions for trajectory prediction.

As a side note, we point out that SLSTM [1] and NRI [14] both of which model interactions are often outperformed by VLSTM which does not model interactions. While surprising at first, we found that this has also been confirmed for SLSTM by prior works, namely, Social GAN [8] which has common co-authors with SLSTM, and also independently by the TrajNet Benchmark paper [3]. We believe that this is because both methods introduce significant noise in the neighborhood of agents: (a) SLSTM does this by aggregating agents' hidden states within discretized bins which can potentially lose significant motion specific information, and (b) NRI infers many spurious edges during variational edge-type inference (also shown by [16]).

## 4.2 Ablations

**Modeling only inertia**: We first remove the velocity correction term $(\Delta v_i^t)$ and only retain the constant velocity estimate (inertia) to show that both intention and interaction modeling are indeed required for accurate prediction. We call this model $\text{FQA}_{inert}$ and Table 2 shows the stark deterioration in performance after the removal of velocity correction term.

**Modeling only inertia and agent intention**: We next drop only the interaction module by setting all attention vectors $a_{i=1:N}$ to 0, while keeping the constant velocity estimate and the intentional motion LSTM (eqs 2,3) intact. The resulting RMSEs shown as $\text{FQA}_{NoIntr}$ in Table 2 capture the severe drop in performance on all datasets, thereby showing that a major chunk of improvement indeed comes from modeling the interactions.

**Removing decision making of FQA**: To demonstrate that the strength of the interaction module comes from FQA's decision making process, we next replaced all sub-modules between the inputs of the FQA module uptil $V_{sr}$ in figure 2 with fully-connected layers with equivalent number of learnable parameters so that responses $V_{sr}$ are directly produced from input features without any fuzzy decisions. We call this variant $\text{FQA}_{NoDec}$ and show the deterioration in performance from loss of decision making in Table 2. It is clear that while $\text{FQA}_{NoDec}$ outperforms $\text{FQA}_{inert}$ and $\text{FQA}_{NoIntr}$ because it models interactions with at least a simple neural network, substituting the decision making mechanism has reduced FQA to the same or worse level of performance as other baselines on most benchmark datasets.

## 4.3 Understanding fuzzy decisions of FQA

**Distance-based cutoff for edges**: To check if FQA can learn decisions to reflect proximity between agents, we replaced our edge generator to produce edges with a distance-based cutoff so it outputs a directed edge between agents $s$ and $r$ only if $\|p_s^t - p_r^t\|_2 \leq d_{thresh}$. The threshold $d_{thresh}$ was found by a crude hyperparameter search and was set to $d_{thresh} = 0.5$ in the normalized coordinates provided to all models. We show prediction errors for FQA and other baselines namely GN, GSAGE

Table 2: Prediction error metrics with ablations and augmentations

| Model | ETH-UCY | Collisions | NGsim | Charges | NBA |
|---|---|---|---|---|---|
| $FQA_{inert}$ | $0.576 \pm 0.000$ | $0.519 \pm 0.000$ | $6.159 \pm 0.000$ | $0.778 \pm 0.000$ | $13.60 \pm 0.000$ |
| $FQA_{NoIntr}$ | $0.549 \pm 0.006$ | $0.236 \pm 0.0003$ | $5.756 \pm 0.152$ | $0.523 \pm 0.001$ | $6.038 \pm 0.044$ |
| $FQA_{NoDec}$ | $0.539 \pm 0.006$ | $0.234 \pm 0.001$ | $5.616 \pm 0.163$ | $0.505 \pm 0.007$ | $5.518 \pm 0.049$ |
| $GN_{dce}$ | $0.572 \pm 0.020$ | $0.227 \pm 0.002$ | $\mathbf{5.714 \pm 0.155}$ | $0.451 \pm 0.004$ | $\mathbf{5.553 \pm 0.010}$ |
| $GSAGE_{dce}$ | $0.579 \pm 0.011$ | $0.231 \pm 0.001$ | $5.901 \pm 0.099$ | $0.456 \pm 0.005$ | $5.898 \pm 0.048$ |
| $GAT_{dce}$ | $0.571 \pm 0.006$ | $0.232 \pm 0.001$ | $5.936 \pm 0.124$ | $0.460 \pm 0.008$ | $5.938 \pm 0.021$ |
| $FQA_{dce}$ | $\mathbf{0.532 \pm 0.002}$ | $\mathbf{0.175 \pm 0.004}$ | $5.814 \pm 0.170$ | $\mathbf{0.416 \pm 0.001}$ | $5.733 \pm 0.033$ |
| $FQA_{hk}$ | $0.541 \pm 0.002$ | $0.177 \pm 0.006$ | $4.801 \pm 0.215$ | $0.396 \pm 0.007$ | $5.457 \pm 0.084$ |

and GAT[6] by providing them distance-constrained edges instead of all edges (*dce* variants) in Table 2. While *dce* variants of baselines show improvement in prediction errors on most datasets, FQA only shows minor improvements on Collisions which has sparse density of interactions, while the performance degrades on the other datasets with dense interactions. This suggests that FQA is indeed able to model proximity between agents even from a fully-connected graph, if the dataset is sufficiently dense in the number of interactions per time-step and does not require aiding heuristics, while other baselines do not necessarily extract this information and hence benefit from the heuristic.

Table 3: Predict collisions from FQA decisions

| $\tau$ | 1 | 2 | 3 | Recurrent |
|---|---|---|---|---|
| Accuracy | 95.55% | 95.48% | 95.35% | 95.75% |
| AUROC | 0.854 | 0.866 | 0.870 | 0.907 |

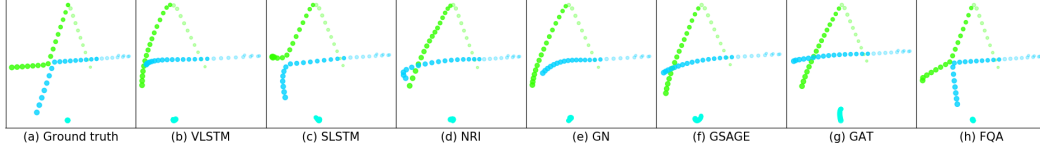

(a) Ground truth  (b) VLSTM  (c) SLSTM  (d) NRI  (e) GN  (f) GSAGE  (g) GAT  (h) FQA

(a) Collisions data: FQA models sparse interactions like inter-agent collisions well.

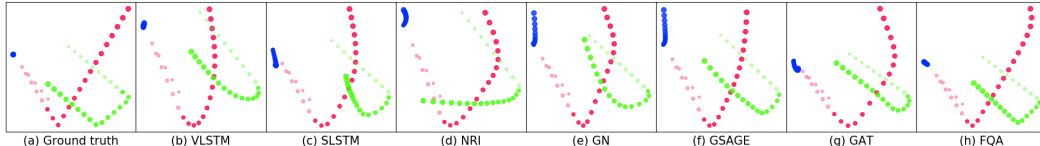

(a) Ground truth  (b) VLSTM  (c) SLSTM  (d) NRI  (e) GN  (f) GSAGE  (g) GAT  (h) FQA

(b) Collsions data: FQA models stationary fixed landmarks well (blue) and predicts sharp collisions with walls.

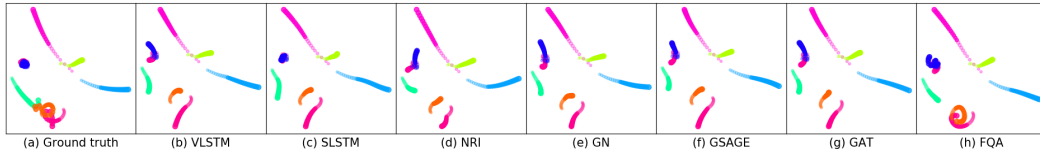

(a) Ground truth  (b) VLSTM  (c) SLSTM  (d) NRI  (e) GN  (f) GSAGE  (g) GAT  (h) FQA

(c) Charges data: Complex swirling in opposite charges (see pink and orange trajectories) accompanied by high accelerations; No model except FQA is able to predict such complex motion.

Figure 3: Predicted trajectories from all models shown with circles of radii increasing with time. The lighter shades show the observed part uptil $T_{obs}$ while the darker shades show the predictions till $T$.

**Predicting interactions from decisions**: To investigate if the decisions capture inter-agent interactions well, we present an experiment to predict when a collision happens between two agents on the

Collisions dataset[7] from only the 8 agent-pair decisions $D_{sr}^t$. Since collisions are sparse, we present the prediction accuracy and the area under the ROC curve on a held-out test set in Table 3 for various classifiers trained to predict collisions between agents using different horizon of time-steps ($\tau$) of the input decisions. Note that we do not even use the agents' positions, velocities or the FQA responses ($V_{sr}$) as inputs to the predictors. Yet, the decision-trajectories alone are sufficient to predict collisions with a surprisingly high accuracy and AUROC, which strongly indicates that FQA's decisions are accurately capturing inter-agent interactions.

**Including human-knowledge in FQA**: Next we show that one can also add fuzzy decisions to FQA, which are intuitive for humans but might be hard to infer from data. To this end, we add an additional fixed decision $D = \sigma(\tilde{v}_{sr}^T \hat{p}_{sr})$ to FQA which should tend to 0 (*no*) when two agents are directly approaching each other, while leaving the corresponding yes-no responses learnable (we call this FQA$_{hk}$). While Table 2 shows no significant improvement on most datasets, presumably since the information captured by this decision is already being captured by the model, we do observe a significant decrease in RMSE on the NGsim dataset compared to Table 1. This is because our chunking procedure on NGsim eliminates a few neighbors of the agents at sub-scene boundaries and consequently certain interaction effects become harder to capture from data. So adding this human-knowledge directly as a decision improves performance. Hence, FQA allows the designer to augment the model with human-knowledge decisions as hints, which can improve performance and are ignored if not useful.

**Visualization**: Next we visualize the trajectories predicted by FQA and other baselines. Figures 3a and 3b show inter-agent collisions and those between agents and boundaries respectively. Due to agents' small sizes, inter-agent collisions are sparse events and only FQA learns to model them appropriately while the other baselines ignore them. Further FQA models the trajectories of agents faithfully and all collisions sharply while other baselines sometimes predict curved trajectories and premature soft collisions in empty space without any real interaction. We further observe from the pink and orange charges in Figure 3c, that it is hard to model chaotic swirling of nearby opposite charges due to high accelerations resulting from coulombic forces and that FQA comes closest to being an accurate model. More visualization examples are shown in the appendix.

**Limitations**: Finally, we point out that FQA (and all baselines) have a high RMSE on the NBA dataset (w.r.t. the relative scale of values in the dataset), which comprises of many sudden intent dependent events or otherwise motions with many valid alternatives that cannot be predicted in the long term[8]. For such datasets, we recommend making shorter length predictions or including visual observations in the input instead of just trajectory data to account better for strong intent-dependencies. Alternatively, FQA being primarily designed to target interactions, can be combined with stronger models for modeling intents, e.g., hierarchical policy networks [30] to improve performance on intent-driven prediction setups. Please see the appendix for a more detailed analysis on the NBA dataset.

# 5   Conclusion

We have presented a general architecture designed to predict trajectories in multi-agent systems while modeling the crucial inductive biases of motion, namely, inertia, relative motion, intents and interactions. Our novel Fuzzy Query Attention (FQA) mechanism models pairwise interactions between agents by learning to make fuzzy (continuous-valued) decisions. We demonstrate significant performance gains over existing state-of-the-art models in diverse domains thereby demonstrating the potential of FQA. We further provide ablations and empirical analysis to understand the strengths and limitations of our approach. FQA additionally allows including human-knowledge in the model by manually inserting known decisions (when available) and learning their corresponding responses. This could be useful for debugging models in practical settings and at times aligning the model's decisions to human expectations.

## Broader Impact

We have presented a general architecture for multi-agent trajectory prediction which includes the crucial inductive biases of motion. Our FQA attention mechanism models interactions in multi-agent trajectory prediction and outperforms existing state-of-the-art models in many diverse settings. Our architecture relies only on trajectory data and hence can be employed in conjunction to or alternatively as part of visual processing pipelines for trajectory prediction. It can be successfully incorporated in deep learning pipelines for predicting traffic trajectories around self-driving autonomous vehicles, predicting motion of pedestrians on roads etc. Note that while FQA is primarily designed to target interactions, it can be combined with stronger models for modeling intents, e.g., hierarchical policy networks [30] to improve performance on intent-driven prediction setups e.g. in sports analytics for predicting valid or alternative strategies for basketball players.

## Acknowledgments and Disclosure of Funding

This research was supported in part by NSF Research Grant IIS-1254206 and MURI Grant W911NF-11-1-0332. Hao Zhu and Ming Zhang were supported by National Key Research and Development Program of China with Grant No. 2018AAA0101900 / 2018AAA0101902 as well as the National Natural Science Foundation of China (NSFC Grant No. 61772039 and No. 91646202).

## Footnotes

[1]We use the word *fuzzy* in this work to represent continuous-valued decisions over their discrete-valued boolean counterparts and not necessarily to fuzzy logic.

[2]We also show experiments with edges based on distance-based cutoffs as previous work [4] has found this heuristic useful for trajectory prediction.

[3]Note that the word *fuzzy* represents continuous-valued decisions over their discrete-valued boolean counterparts and not fuzzy logic.

[4]The detach operator acts as identity for the forward-pass but prevents any gradients from propagating back through its operand. This allows us to learn feature representations only using responses while the keys and queries make useful decisions from the learnt features.

[5]Code for implementing FQA can be found at https://github.com/nitinkamra1992/FQA.git

[6]SLSTM already uses a neighborhood size of $0.5$ for discretization, while NRI infers edges internally via variational inference.

[7]This is the only synthetic dataset for which the ground truth of interactions is available.

[8]Note that FQA is still the most accurate trajectory predictor amongst our baselines on the NBA dataset.

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
