[Supplementary Material]

# A   Appendix

## A.1   Model architectures and hyperparameters

We provide the hyperparameters of our baselines and those of FQA in this section. All our experiments were done on systems with Ubuntu 16.04 and all models trained using either Nvidia Titan X or Nvidia GeForce GTX 1080 Ti GPUs. All code was written in Python 3.6 with neural network architectures defined and trained using PyTorch v1.0.0. The code for implementing FQA can be found at https://github.com/nitinkamra1992/FQA.git.

### A.1.1   Vanilla LSTM

The Vanilla LSTM model embeds each $p_i^t$ to a 32-dimensional embedding vector using a fully-connected layer with ReLU activation. This vector is fed along with the previous hidden states to an LSTM with state size 64, whose output is again processed by a fully-connected layer to generate the 2-dimensional offset for next-step prediction.

### A.1.2   Social LSTM

We adapted the code from `https://github.com/quancore/social-lstm` which directly reproduces the original authors' model from [1]. We kept the initial embedding size as 20, the LSTM's hiddden size as 40, the size of the discretization grid as 4 and the discretization neighborhood size as $0.5^9$.

### A.1.3   Neural Relational Inference

We adapted the authors' official repository from `https://github.com/ethanfetaya/NRI`. The input dimension was kept as 2 for positional coordinates and the number of edge types as 3 (since setting it to 2 gave worse results). The encoder employed the MLP architecture with hidden layers of sizes 32 and no dropout, while the GRU-based RNN architecture was used for the decoder with hidden state of size 32 and no dropout. The variance of the output distribution was set to $5 \times 10^{-5}$.

### A.1.4   Graph Networks

While the original repository for Graph Networks is written in TensorFlow (`https://github.com/deepmind/graph_nets`), we translated the repository into PyTorch and adapted models similar to those employed by [23, 2]. We employed a vertex-level encoder followed by a recurrent Graph Network based on GRU-style recurrence followed by a Graph Net decoder. The vertex-level encoder transforms 2-dimensional positional input at each time step to a 10-dimensional node embedding. An input graph is constructed from these node embeddings having all pairwise edges and dimensions 10, 1 and 1 respectively for the node, edge and global attributes. This input graph along with a previous state graph (with dimensions 45, 8 and 8 for node, edge and global state attributes) was processed using a GRU-style recurrent Graph Network to output the updated state graph of the same dimensions (45, 8 and 8 for node, edge and global state attributes respectively). This new state graph was processed by a feedforward graph-network as prescribed in [2] to output another graph whose node features of dimensions 2 were treated as offsets for the next time step prediction. All update networks both in the encoder and the decoder (for node, edge and global features) used two feedforward layers with the intermediate layer having latent dimension 32 and a ReLU activation. While the original work proposes to use sum as the aggregation operator, we found summing to often cause the training to divergence since different agents have neighborhoods of very diverse sizes ranging from 0 to about 40 at different times in many of our datasets. Hence we used feature-wise mean-pooling for all aggregation operators.

### A.1.5   GraphSAGE, Graph Attention Networks and Fuzzy Query Attention

Since GraphSAGE (GSAGE) [9] and Graph Attention Networks (GAT) [26] were not originally prescribed for a multi-agent trajectory prediction application, we used their update and aggregation styles in our own FQA framework to replace the FQA sub-module in our Interaction module described

in Section 3. For all three methods the input size and the output size was 2, while the hidden state dimension of the LSTM shared by all agents was 32. The dimension of the aggregated attention for each agent $a_i^t$ was also set to 32 for all three methods. All the three methods involved the $FC_1, FC_2, FC_3$ and $FC_4$ layers described in section 3 and had the output sizes $48, 32, 16$ and $2$ respectively.

**GSAGE**: GraphSAGE [9] directly embeds all sender latent vectors $h_s$ into 32-dimensional embeddings via two fully-connected layers each with a RELU activation and with the intermediate layer of dimensions 32. The output embeddings were aggregated into the receiver nodes via feature-wise max-pooling to generate $a_i^t$.

**GAT**: GAT performs a similar embedding of sender hidden states using a similar embedding network as GSAGE but aggregates them via feature-wise max-pooling after weighing the embeddings with 8 attention head coefficients generated as proposed in [26] and finally averages over the 8 aggregations. We used 8 attention heads to match the number of FQA's decisions.

**FQA**: FQA used 8 query-key pairs for all datasets leading to 8 decisions. The dimension for keys and queries was set to 4, while the dimension for yes-no responses was kept as 6. Consequently the dimension of learnt bias vector $B$ was also 8 and the sizes of the fully-connected layers $FC_5, FC_6, FC_7, FC_8, FC_9, FC_{10}, FC_{11}$ and $FC_{12}$ described in the main text were $32, 32, 33, 48, 33, 48, 32$ and $32$ respectively.

## A.2 Limitations

With visualizations on the NBA dataset we highlight when our setup and most interaction modeling approaches may not be useful for trajectory prediction. Figure 4 shows a scene from the NBA dataset with the ball trajectory being green and the team players being blue and red trajectories. A blue player carries the ball and passes it to a teammate at the corner of the field after the observation period $(2T/5)$ ends, which turns all the red player trajectories towards that corner (ground truth). Such passes and consequent player motions are heavily intent dependent and quite unpredictable. Most methods e.g. FQA instead predicate an equally valid alternative in which the original blue player carries the ball towards the basket. NBA dataset comprises of many such intent dependent sudden events or otherwise motions with many valid alternatives which cannot be predicted in the long term $(3T/5)$. For such datasets, we recommend making shorter length predictions or including visual observations for making predictions instead of just trajectory data. Note that FQA is still the most accurate trajectory predictor even on this dataset. Figure 5 shows three other cases where a player chooses to counter-intuitively pass (or not pass) the ball after the observation period ends. Most methods, especially FQA, predict an equally valid and often more likely alternative of not passing the ball or passing it in a direction more logically deducible from only trajectory data.

(a) Ground truth    (b) VLSTM    (c) SLSTM    (d) NRI    (e) GN    (f) GSAGE    (g) GAT    (h) FQA

Figure 4: NBA data: Green agent is the ball, while the 5 players in each team are colored blue and red. The pass between blue team players is unpredictable and heavily intention dependent.

## A.3 Additional visualizations

Next we show additional visualization from all models on all datasets (other than NBA). The visualizations clearly demonstrate the strong inductive bias of FQA for multi-agent trajectory prediction.

Figure 5: Predicted trajectory visualization from various models on the NBA dataset.

Figure 6: Predicted trajectory visualization from various models on Charges dataset.

Figure 7: Predicted trajectory visualization from various models on ETH-UCY dataset.

Figure 8: Predicted trajectory visualization from various models on Collisions dataset.

Figure 9: Predicted trajectory visualization from various models on NGsim dataset.

## Footnotes

[9]This neighborhood size is also the same as the distance cutoff used in section 4.3.