[Reviews · NeurIPS 2020]

Review 1

Summary and Contributions: The paper proposes a novel attention block termed as ‘Fuzzy Query Attention’ (FQA) to better model social interactions between interacting agents. The crux of the FQA block lies in learning decision variables which dictate how the human reacts to his/her neighbor. In particular, the model learns how to react in a yes/no (binary) scenario (eg. possibility of collision) respectively and the decision variable indicates the probability of that particular scenario occurring. The proposed FQA has been shown mathematically to have the potential to model important features like proximity, approach. The experimentation has been extensive, and the overall model is shown to outperform related works on multiple datasets.

Strengths: The paper proposes a novel attention module to better capture social interactions. The architecture of the module, with regards to two-agent interactions, matches the human intuition of decision making. The authors further support it by demonstrating, mathematically, two common decisions of Proximity and Approach. The experiments have been extensive with regards to different datasets chosen, and they strongly indicate the effectiveness of the proposed architecture. Both quantitatively and quantitively, the performance of the proposed model is strong. I personally like the diversity of the datasets, from human walking dynamics and vehicles and physics data. The paper is very well-written and has been enjoyable to read.

Weaknesses: The experiments have been extensive, however I have following three crucial questions to better understand the performance boost arising from the overall architecture: 1. Improvement arising from interaction module or motion module? Taking Social LSTM [1] to be an interaction-based baseline, the proposed architecture has two different components: the interaction and motion modules. Is the boost coming from the interaction module which is FQA in comparison to Social Pooling [1]? Or is it the new motion module? An ablation study showing the performance while keeping the motion module the same as the baseline will help answer this question. 2. Are the decisions actually Fuzzy? The authors use the term Fuzzy to describe continuous-valued decisions over their discrete-valued boolean counterparts. It is therefore important to ask the question what happens within the network if the Di’s were Boolean. What is the performance drop? 3. Comparison to self-attention [2]: The authors provide an ablation study ‘Removing decision making of FQA’. It is slightly confusing as to what exactly is the architecture without decision-making? Do the authors perform a procedure similar to self-attention (use K, Q and V) to determine Vsr. If no, it would be interesting to see the comparison with self-attention block. [1] Social LSTM: Human Trajectory Prediction in Crowded Spaces, 2016 [2] Attention is all you need, 2016

Correctness: The methodology is correct.

Clarity: Yes the paper is well written

Relation to Prior Work: yes, it is clearly discussed

Reproducibility: Yes

Additional Feedback: - In the rebuttal, it will be great if the authors could answer the questions in the "Weakness section". - It would be further good to see comparison to non-graph based trajectory models in ETH-UCY. - As a suggestion, maybe use the term ‘gating mechanism’ as opposed to ‘fuzzy’. POST REBUTTAL: Thank you for the clarifications. I have updated my score.


Review 2

Summary and Contributions: The authors propose Fuzzy Query Attention (FQA), a pair-wise attention mechanism for relational models. Similar to Graph Nets or Transformers, FQA captures the effects of the interaction of a sender node on a receiver node with a function of the pair. In particular, the function of choice is a "fuzzy" dot product that allows the network to interpolate between the presence and absence of an effect.

Strengths: The paper is well written and the ideas presented are appropriately placed in the context of existing work. The work is certainly relevant to the NeurIPS community: relational reasoning has received considerable attention in recent years. The ideas are novel and results are impressive and simple to understand.

Weaknesses: This is more a suggestion than a weakness, it would be nice to see how this methods scale up to very large systems (e.g. the simulations in Learning to simulate complex physics with graph networks). Possibly FQA might drop edges from large graphs in a context-dependent way, and might be able to do so dynamically. It would be interesting to see if it helps. Similarly, it would be interesting to have a look at a thresholded version of these fuzzy decisions on over-connected graphs. Do they correctly recover the tru underlying structure?

Correctness: Yes

Clarity: Yes

Relation to Prior Work: Yes

Reproducibility: Yes

Additional Feedback: Thank you for sharing these cool ideas! I have two random thoughts for future directions I thought I might share. 1. If I understand the math correctly, there is nothing limiting your method to being fuzzy between only 2 values. In particular, you could produce a set of queries and keys with a multi-headed network, and then construct a few decisions D_sr,1; D_sr,2;...;D_sr,m, pass them through a soft-max (as if they were logits), and construct a linear combination of V_1, V_2, ..., V_m, which can be produced similarly to V_y and V_n. This would be especially useful in the presence of a mixed system (e.g. a mass-spring chain falling on a rigid surface, you'd have the mass-spring interaction, as well as the mass-surface). 2. This is minor, I wonder if it would be wise to normalize V_y and V_n to have norm 1 or something. I am worried unbounded values might interfere with your fuzzy decisions (e.g. if \|V_y\| >> \|V_n\| your fuzzy decision is not working as intended). I like this research direction, I hope my feedback helps. All the best. ============ After Author Feedback =========== Thank you for taking the time to write a rebuttal, and for sharing these cool ideas. I stand by my initial assessment that this paper should be accepted. Best.


Review 3

Summary and Contributions: This work presents a novel attention mechanism for modeling edge interactions in a graph. It is demonstrated by modeling physical simulations as well as real-world pedestrian and vehicle interactions.

Strengths: The experimental evaluation is quite thorough, with a good ablation study demonstrating where performance comes from.

Weaknesses: A lack of clarity in Section 3 really hinders understanding how the method works. Notably, how can the FC layers behind K_sr and Q_sr be trained if their output ends up being detached? Why are the keys and queries necessary? What do they do? If the FC layers cannot be trained, then aren't K and Q essentially just random vectors? To remedy this, an overview paragraph stating something like "At a high level, FQA performs the following ... FQA generates keys K and queries Q that represent ... The decision variables D then select the most relevant parts of K and Q to ..." would really aid a reader's understanding. This set of questions: "How can the FC layers behind K_sr and Q_sr be trained if their output ends up being detached? Why are the keys and queries necessary? What do they do? If the FC layers cannot be trained, then aren't K and Q essentially just random vectors?" are my main identified weaknesses with the work, and are the reason behind the initial review score. If these questions are answered satisfactorily, I am happy to increase my score to be above the acceptance threshold. POST-REBUTTAL: Thank you for the clarification! My questions were answered and thus I will raise my score accordingly. Good luck!

Correctness: Yes, the experimental methodology follows existing work and claims in the paper are substantiated by the experiments.

Clarity: The paper's general grammar is fine, but Section 3 was hard to follow at times and I found it difficult to parse out what exactly is going on in the model.

Relation to Prior Work: This could be done a little bit better. In particular, what concretely separates FQA from existing attention architectures? Yes, it is evident that this fuzzy combination of keys and queries is different, but it is hard to find any deeper insight as to why this is intuitively a better approach or how different this really is compared to common dot-product attention methods (e.g., Bahdanau attention)? Such information can be put into lines 128-138, which is where I was hoping to find such a discussion.

Reproducibility: Yes

Additional Feedback: Line 114: "parallely parsed" -> "parsed in parallel"


Review 4

Summary and Contributions: The authors propose a method for multi-agent trajectory prediction. They present a relational model to describe interaction between agents with a fuzzy query attention mechanism. The method is evaluated in some datasets such as human crowd trajectories, US freeway traffic, NBA sports data, and show its validity in different environments.

Strengths: Modeling interaction is important in trajectory prediction. FAQ module proposed in this paper looks good. Experiments are sufficient and show that the proposed FAQ module has high validity. The paper is well written and the mathematical framework seems correct.

Weaknesses: There are a few points I'd like to discuss. 1. Making continuous-valued decision, especially continuous attention between agents, is not new in trajectory prediction. However, FAQ module proposed in this paper looks new. 2. The decision demonstrated in section 4.3 looks too simple, which could be easily evolved by other methods. Overall, I think the innovation of this paper is limited.

Correctness: The method seems corrent.

Clarity: Well written.

Relation to Prior Work: Clearly discussed.

Reproducibility: Yes

Additional Feedback:

[Author Response · NeurIPS 2020]

We heartily thank all the reviewers for their thoughtful comments and valuable feedback. We will try to address your
queries and clarify any misunderstandings below.

**To R1**:

1. We have not used the word "motion module" in the paper, so it is slightly unclear which sub-module you are refering
to. From the comments, our best guess is that you are referring to the linear velocity estimator (eq 2) and the intentional
motion LSTM (eq 3) jointly as motion module. To highlight the improvement caused by the FQA interaction module,
we show another ablation dropping the interaction module completely by setting all attention vectors to $0$, while keeping
the linear velocity estimate and the intentional motion LSTM (eqs 2, 3) intact. The resulting RMSEs shown below
capture the severe drop in performance on all datasets, thereby showing that the improvement indeed comes from the
interaction module.

| ETH-UCY | Collisions | NGsim | Charges | NBA |
|---|---|---|---|---|
| $0.549 \pm 0.006$ | $0.236 \pm 0.0003$ | $5.756 \pm 0.152$ | $0.523 \pm 0.001$ | $6.038 \pm 0.044$ |

2. Having binary decisions would make the architecture non-differentiable. Alternatively one maintains bernoulli
decision RVs and infers their distributions using variational inference while making predictions. This is exactly what
our NRI baseline does (and it is outperformed by FQA).

3. When removing the "decision making", we replace all sub-modules between the inputs of FQA module uptil $V_{sr}$
(fig 2 in paper) and replace them with fully-connected layers with equivalent number of learnable params. A direct
comparison with self-attention is not feasible since self-attention acts on a single entity, not a pair of entities (s and r).
In fact, one way to interpret our FQA is as extending self-attention to pairs of entities.

**To R2**:

Thank you for the excellent suggestions! Indeed FQA can be easily extended to being fuzzy between multiple values
which could be beneficial when multiple mixed decisions are required. We also found the "Learning to simulate complex
physics with graph networks" really cool and relevant. It might be hard to implement it right away given the limited
rebuttal time, but we will be happy to cite it in the final draft.

**To R3**:

We want to clarify a misunderstanding here. We do not detach the keys and queries ($K_{sr}$ and $Q_{sr}$), else as you rightly
mentioned, the FC layers preceding them would end up being random vectors. We detach the copy of features $f_{sr}$
entering into the fully-connected layers which produce the keys and queries. This is also mentioned in lines $135 - 136$
and mathematically expressed in eqs 8 and 9. Note that gradients still backpropagate into $f_{sr}$ via the yes-no responses:
$V_{y,sr}$ and $V_{n,sr}$ since they do not received detached features. This way all layers have gradients backpropagated into
them and no layer is left hanging.

While we have provided explanations for most quantities as they are introduced in section 3, we understand that it
can be terse to parse the details in bits and pieces. We will remedy this with your suggestion and precede the detailed
description with an overview paragraph explaining the architecture at a high level in the final draft. We will also expand
our description of broader impact of our work and will be happy to take suggestions if you would like to offer some.

**To R4**:

The example in lines 19-22 is a motivational example for why continuous-valued decisions are required in decision
making. We do not (over)-claim that our architecture will learn to make such human interpretable decisions.

In fact, there is no concrete way of forcing neural nets to learn human-interpretable representations on their own just by
observing trajectory data without providing any additional human knowledge and we do not claim to do so (we explicitly
mention this in lines 149-150). For instance, if quantities $x$ and $y$ are interpretable for humans, a neural net might easily
learn other quantities, e.g. $x + y$ and $x - y$, which might be meaningless for humans but contain the same information
as $x$ and $y$. One can only check if our learnt decisions contain enough information about human-intepretable quantities
and we have shown an instance of this in section 4.3 by predicting collisions solely from our learnt decisions (lines
$245 - 253$).

Additionally, our architecture does allow providing pre-defined decisions which could come from human-knowledge
and lets the model learn appropriate responses to them. This is not a trivial feature found in existing methods and is
very useful in practical settings for debugging and for aligning the model to human expectations to some extent.

[Meta-Review · NeurIPS 2020]

summary: The authors propose an improved neural network architecture for predicting time series representing pairwise interacting entities. The architecture is based on a soft attention mechanism over pairwise relations among entities. pros: - addresses important challenge of relational reasoning - demonstrates improved empirical performance over relevant baselines on a quite wide range of datasets - ablation demonstrating the importance of individual model parts cons: - somewhat opaque presentation meta review: Paper with good empirical results on a relevant topic. Accept.